# Lexically Constrained Decoding for Sequence Generation Using Grid Beam Search

## Abstract

We present Grid Beam Search (GBS), an algorithm which extends beam search to allow the inclusion of pre-specified lexical constraints. The algorithm can be used with any model that generates a sequence $\hat{y} = \{y_0 \ldots y_T\}$, by maximizing $p(\mathbf{y}|\mathbf{x}) = \prod_t p(y_t|\mathbf{x}; \{y_0 \ldots y_{t-1}\})$. Lexical constraints take the form of phrases or words that must be present in the output sequence. This is a very general way to incorporate additional knowledge into a model's output without requiring any modification of the model parameters or training data. We demonstrate the feasibility and flexibility of Lexically Constrained Decoding by conducting experiments on Neural Interactive-Predictive Translation, as well as Domain Adaptation for Neural Machine Translation. Experiments show that GBS can provide large improvements in translation quality in interactive scenarios, and that, even without any user input, GBS can be used to achieve significant gains in performance in domain adaptation scenarios.

## 1 Introduction

The output of many natural language processing models is a sequence of text. Examples include automatic summarization (Rush et al., 2015), machine translation (Koehn, 2010; Bahdanau et al., 2014), caption generation (Xu et al., 2015), and dialog generation (Serban et al., 2016), among others.

In many real-world scenarios, additional information that could inform the search for the optimal output sequence may be available at inference time. Humans can provide corrections after viewing a system's initial output, or separate classification models may be able to predict parts of the output with high confidence. When the domain of the input is known, a domain terminology may be employed to ensure specific phrases are present in a system's predictions. Our goal in this work is to find a way to force the output of a model to contain such *lexical constraints*, while still taking advantage of the distribution learned from training data.

For Machine Translation (MT) usecases in particular, final translations are often produced by combining automatically translated output with user inputs. Examples include Post-Editing (PE) (Koehn, 2009; Specia, 2011) and Interactive-Predictive MT (Foster, 2002; Barrachina et al., 2009; Green, 2014). These interactive scenarios can be unified by considering user inputs to be lexical constraints which guide the search for the optimal output sequence.

In this paper, we formalize the notion of lexical constraints, and propose a decoding algorithm which allows the specification of subsequences that are required to be present in a model's output. Individual constraints may be single tokens or multi-word phrases, and any number of constraints may be specified simultaneously.

Although we focus upon interactive applications for MT in our experiments, lexically constrained decoding is relevant to any scenario where a model is asked to generate a sequence $\hat{y} = \{y_0 \ldots y_T\}$ given both an input $\mathbf{x}$, and a set $\{\mathbf{c_0} ... \mathbf{c_n}\}$, where each $\mathbf{c_i}$ is a sub-sequence $\{c_{i0} \ldots c_{ij}\}$, that must appear somewhere in $\hat{y}$. This makes our work applicable in a wide range of text generation scenarios, including image description, dialog generation, automatic summarization, and question answering.

The rest of this paper is organized as follows: Section 2 gives the necessary background for our

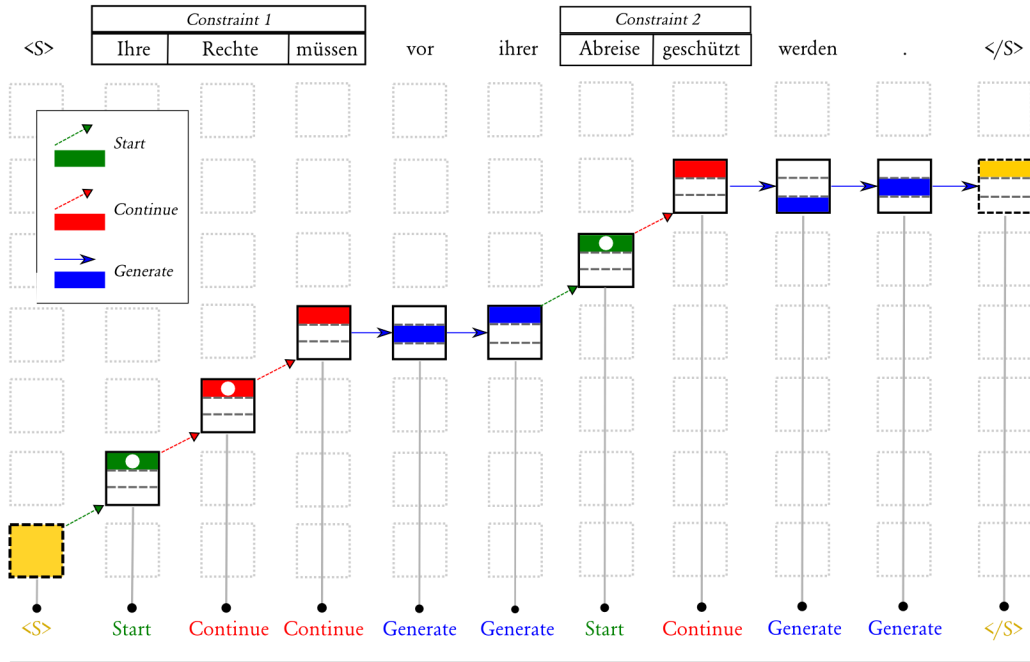

Input: Rights protection should begin before their departure .

Figure 1: A visualization of the decoding process for an actual example from our English-German MT experiments. The output token at each timestep appears at the top of the figure, with lexical constraints enclosed in boxes. *Generation* is shown in blue, *Starting* new constraints in green, and *Continuing* constraints in red. The function used to create the hypothesis at each timestep is written at the bottom. Each box in the grid represents a beam; a colored strip inside a beam represents an individual hypothesis in the beam's $k$-best stack. Hypotheses with circles inside them are *closed*, all other hypotheses are *open*. (Best viewed in colour).

discussion of GBS, Section 3 discusses the lexically constrained decoding algorithm in detail, Section 4 presents our experiments, and Section 5 gives an overview of closely related work.

## 2 Background: Beam Search for Sequence Generation

Under a model parameterized by $\theta$, let the best output sequence $\hat{\mathbf{y}}$ given input $\mathbf{x}$ be Eq. 1.

$$\hat{\mathbf{y}} = \underset{\mathbf{y} \in \{\mathbf{y}^{[\mathbf{T}]}\}}{\operatorname{argmax}} p_\theta(\mathbf{y}|\mathbf{x}), \qquad (1)$$

where we use $\{\mathbf{y}^{[\mathbf{T}]}\}$ to denote the set of all sequences of length $T$. Because the number of possible sequences for such a model is $|\mathbf{v}|^T$, where $|\mathbf{v}|$ is the number of output symbols, the search for $\hat{\mathbf{y}}$ can be made more tractable by factorizing $p_\theta(\mathbf{y}|\mathbf{x})$ into Eq. 2:

$$p_\theta(\mathbf{y}|\mathbf{x}) = \prod_{t=0}^{T} p_\theta(y_t|\mathbf{x}; \{y_0 \dots y_{t-1}\}). \qquad (2)$$

The standard approach is thus to generate the output sequence from beginning to end, conditioning the output at each timestep upon the input $\mathbf{x}$, and the already-generated symbols $\{y_0 \dots y_{i-t}\}$. However, greedy selection of the most probable output at each timestep, i.e.:

$$\hat{y}_t = \underset{y_i \in \{\mathbf{v}\}}{\operatorname{argmax}} p(y_i|\mathbf{x}; \{y_0 \dots y_{t-1}\}), \qquad (3)$$

risks making locally optimal decisions which are actually globally sub-optimal. On the other hand, an exhaustive exploration of the output space would require scoring $|\mathbf{v}|^T$ sequences, which is intractable for most real-world models. Thus, a search or *decoding* algorithm is often used as a compromise between these two extremes. A common solution is use a heuristic search to attempt

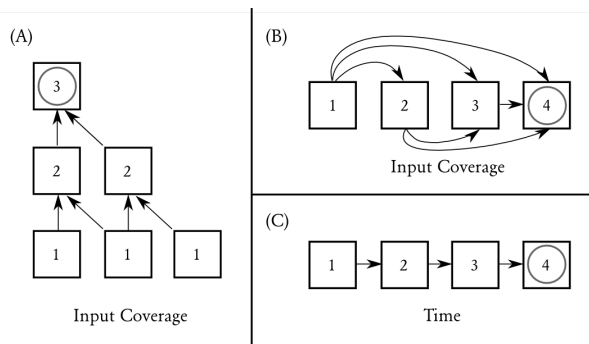

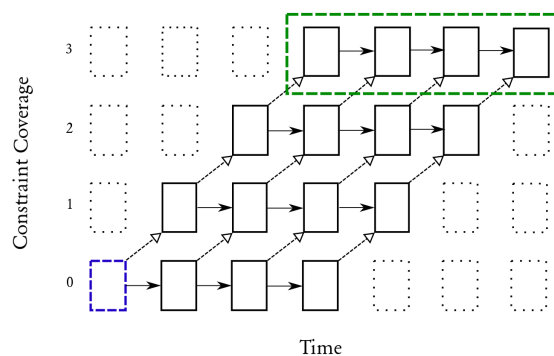

Figure 2: Different structures for beam search. Boxes represent beams which hold $k$-best lists of hypotheses. (A) Chart Parsing using SCFG rules to cover spans in the input. (B) Source coverage as used in PB-SMT. (C) Sequence timesteps (as used in Neural Sequence Models), GBS is an extension of (C). In (A) and (B), hypotheses are finished once they reach the final beam. In (C), a hypothesis is only complete if it has generated an end-of-sequence (EOS) symbol.

Figure 3: Visualizing the lexically constrained decoder's complete search graph. Each rectangle represents a beam containing $k$ hypotheses. Dashed (diagonal) edges indicate *starting* or *continuing* constraints. Horizontal edges represent *generating* from the model's distribution. The horizontal axis covers the timesteps in the output sequence, and the vertical axis covers the constraint tokens (one row for each token in each constraint). Beams on the top level of the grid contain hypotheses which cover all constraints.

to find the best output efficiently (Pearl, 1984; Koehn, 2010; Rush et al., 2013). The key idea is to discard bad options early, while avoiding discarding candidates that may be locally risky, but could eventually result in the best overall output.

Beam search (Och and Ney, 2004) is probably the most popular search algorithm for decoding sequences. Beam search is simple to implement, and is flexible in the sense that the semantics of the graph of beams can be adapted to take advantage of additional structure that may be available for specific tasks. For example, in Phrase-Based Statistical MT (PB-SMT) (Koehn, 2010), beams are organized by the number of source words that are covered by the hypotheses in the beam – a hypothesis is "finished" when it has covered all source words. In chart-based decoding algorithms such as CYK, beams are also tied to coverage of the input, but are organized as cells in a chart, which facilitates search for the optimal latent structure of the output (Chiang, 2007). Figure 2 visualizes three common ways to structure search. (A) and (B) depend upon explicit structural information between the input and output, (C) only assumes that the output is a sequence where later symbols depend upon earlier ones. Note also that (C) corresponds exactly to the bottom rows of Figures 1 and 3.

With the recent success of neural models for text generation, beam search has become the de-facto choice for decoding optimal output sequences (Sutskever et al., 2014). However, with neural sequence models, we cannot organize beams by their explicit coverage of the input. A simpler alternative is to organize beams by output timesteps from $t_0 \cdots t_N$, where $N$ is a hyperparameter that can be set heuristically, for example by multiplying a factor with the length of the input to make an educated guess about the maximum length of the output (Sutskever et al., 2014). Output sequences are generally considered complete once a special "end-of-sentence"(EOS) token has been generated. Beam size in these models is also typically kept small, and recent work has shown that the performance of some architectures can actually degrade with larger beam size (Tu et al., 2016).

## 3 Grid Beam Search

Our goal is to organize decoding in such a way that we can constrain the search space to outputs which contain one or more pre-specified sub-sequences. We thus wish to use a model's distribution both to "place" lexical constraints correctly, and to generate the parts of the output which are not covered by the constraints.

---

**Algorithm 1** Pseudo-code for Grid Beam Search, note that $t$ and $c$ indices are 0-based

1: **procedure** CONSTRAINEDSEARCH($model, input, constraints, maxLen, numC, k$)
2: **startHyp** $\Leftarrow$ model.getStartHyp($input, constraints$)
3: **Grid** $\Leftarrow$ initGrid($maxLen, numC, k$) ▷ initialize beams in grid
4: **Grid**$[0][0] = $ **startHyp**
5: **for** $t = 1, \ t{+}{+}, \ t < maxLen$ **do**
6: **for** $c = max(0, (numC + t) - maxLen), \ c{+}{+}, \ c \leq min(t, numC)$ **do**
7: $n, s, g = \varnothing$
8: **for each** $hyp \in$ **Grid**$[t-1][c]$ **do**
9: **if** $hyp.isOpen()$ **then**
10: $g \Leftarrow g \bigcup$ model.generate($hyp, input, constraints$) ▷ generate new open hyps
11: **end if**
12: **end for**
13: **if** $c > 0$ **then**
14: **for each** $hyp \in$ **Grid**$[t-1][c-1]$ **do**
15: **if** $hyp.isOpen()$ **then**
16: $n \Leftarrow n \bigcup$ model.start($hyp, input, constraints$) ▷ start new constrained hyps
17: **else**
18: $s \Leftarrow s \bigcup$ model.continue($hyp, input, constraints$) ▷ continue unfinished
19: **end if**
20: **end for**
21: **end if**
22: **Grid**$[t][c] = $ k-argmax $_{h \in n \bigcup s \bigcup g}$ model.score($h$) ▷ k-best scoring hypotheses stay on the beam
23: **end for**
24: **end for**
25: $topLevelHyps \Leftarrow$ **Grid**$[:][numC]$ ▷ get hyps in top-level beams
26: $finishedHyps \Leftarrow hasEOS(topLevelHyps)$ ▷ finished hyps have generated the EOS token
27: $bestHyp \Leftarrow \underset{h \in finishedHyps}{\text{argmax}}$ model.score($h$)
28: **return** $bestHyp$
29: **end procedure**

---

Algorithm 1 presents the pseudo-code for lexically constrained decoding, see Figures 1 and 3 for visualizations of the search process. Beams in the grid are indexed by $t$ and $c$. The $t$ variable tracks the timestep of the search, while the $c$ variable indicates how many constraint tokens are covered by the hypotheses in the current beam. Note that each step of $c$ covers a single constraint $token$. In other words, $constraints$ is an array of sequences, where individual tokens can be indexed as $constraints_{ij}$, i.e. $token_j$ in $constraint_i$. The $numC$ parameter in Algorithm 1 represents the total number of tokens in all constraints.

The hypotheses in a beam can be separated into two types (see lines 9-11 and 15-19 of Algorithm 1):

1. *open* hypotheses can either generate from the model's distribution, or start available constraints,

2. *closed* hypotheses can only generate the next token for in a currently unfinished constraint.

At each step of the search the beam at **Grid**$[t][c]$ is filled with candidates which may be created in three ways:

1. the **open** hypotheses in the beam to the left (**Grid**$[t-1][c]$) may *generate* continuations from the model's distribution $p_\theta(y_i | \mathbf{x}, \{y_0 \dots y_{i-1}\})$,

2. the **open** hypotheses in the beam to the left and below (**Grid**$[t-1][c-1]$) may *start* new constraints,

3. the **closed** hypotheses in the beam to the left and below (**Grid**$[t-1][c-1]$) may *continue* new constraints.

Therefore, the **model** in Algorithm 1 implements an interface with three functions: $generate$, $start$, and $continue$, which build new hypotheses in each of the three ways. Note that the scoring function of the model does not need to be aware of the existence of constraints, but it may be, for example via a feature which indicates if a hypothesis is part of a constraint or not.

The beams at the top level of the grid (beams where $c = numConstraints$) contain hypotheses which cover all of the constraints. Once a hypothesis on the top level generates the EOS token, it can be added to the set of finished hypotheses. The highest scoring hypothesis in the set of finished hypotheses is the best sequence which covers all constraints.

### 3.1 Multi-token Constraints

By distinguishing between **open** and **closed** hypotheses, we can allow for arbitrary multi-token phrases in the search. Thus, the set of constraints for a particular output may include both individual tokens and phrases. Each hypothesis also maintains a coverage vector ensuring that constraints cannot be repeated in a search path – hypotheses which have already covered $constraint_i$ can only $generate$, or $start$ constraints that have not yet been covered.

Note also that discontinuous lexical constraints, such as phrasal verbs in English or German, are easy to incorporate into GBS, by adding *filters* to the search, which require that one or more conditions must be met before a constraint can be used. For example, adding the phrasal verb "ask ⟨someone⟩ out" as a constraint would mean using "ask" as $constraint_0$ and "out" as $constraint_1$, with two filters: one requiring that $constraint_1$ cannot be used before $constraint_0$, and another requiring that there must be at least one $generated$ token between the constraints.

### 3.2 Subword Units

Both the computation of the score for a hypothesis, and the granularity of the tokens (character, subword, word, etc...) are left to the underlying model. Because our decoder can handle arbitrary constraints, there is a risk that constraints will contain tokens that were never observed in the training data, and thus are unknown by the model. Especially in domain adaptation scenarios, some user-specified constraints are very likely to contain unseen tokens. Subword representations provide an elegant way to circumvent this problem, by breaking unknown or rare tokens into character n-grams which are part of the model's vocabulary (Sennrich et al., 2016; Wu et al., 2016). In the experiments in Section 4, we use this technique to ensure that no input tokens are unknown, even if a constraint contains words which never appeared in the training data.

### 3.3 Efficiency

Because the number of beams is multiplied by the number of constraints, the runtime complexity of a naive implementation of GBS is $\mathcal{O}(ktc)$. Standard time-based beam search is $\mathcal{O}(kt)$; therefore, some consideration must be given to the efficiency of this algorithm. Note that the beams in each column $c$ of Figure 3 are independent, meaning that GBS can be parallelized to allow all beams at each timestep to be filled simultaneously. Also, we find that the most time is spent computing the states for the hypothesis candidates, so by keeping the beam size small, we can make GBS significantly faster.

### 3.4 Models

The models used for our experiments are state-of-the-art Neural Machine Translation (NMT) systems using our own implementation of NMT with attention over the source sequence (Bahdanau et al., 2014). We used Blocks and Fuel to implement our NMT models (van Merrinboer et al., 2015). To conduct the experiments in the following section, we trained baseline translation models for English–German (EN-DE), English–French (EN-FR), and English–Portuguese (EN-PT). We created a shared subword representation for each language pair by extracting a vocabulary of 80000 symbols from the concatenated source and target data. See the Appendix for more details on our training data and hyperparameter configuration for each language pair. The $beamSize$ parameter is set to 10 for all experiments.

Because our experiments use NMT models, we can now be more explicit about the implementations of the $generate$, $start$, and $continue$ functions for this GBS instantiation. For an NMT model at timestep $t$, $generate(hyp_{t-1})$ first computes a vector of output probabilities $\mathbf{o}_t = softmax(g(y_{t-1}, s_i, c_i))$[1] using the state information available from $hyp_{t-1}$. and returns the best $k$ continuations, i.e. Eq. 4:

---

[1] we use the notation for the $g$ function from Bahdanau et al. (2014)

$$\mathbf{g}_t = \text{k-argmax}_i \mathbf{o}_{ti}. \qquad (4)$$

The *start* and *continue* functions simply index into the softmax output of the model, selecting specific tokens instead of doing a k-argmax over the entire target language vocabulary. For example, to *start* constraint $\mathbf{c}_i$, we find the score of token $\mathbf{c}_{i0}$, i.e. $\mathbf{o}_{tc_{i0}}$.

## 4 Experiments

### 4.1 Pick-Revise for Interactive Post Editing

Pick-Revise is an interaction cycle for MT Post-Editing proposed by Cheng et al. (2016). Starting with the original translation hypothesis, a (simulated) user first picks a part of the hypothesis which is incorrect, and then provides the correct translation for that portion of the output. The user-provided correction is then used as a constraint for the next decoding cycle. The Pick-Revise process can be repeated as many times as necessary, with a new constraint being added at each cycle.

We modify the experiments of Cheng et al. (2016) slightly, and assume that the user only provides sequences of up to three words which are missing from the hypothesis[2]. To simulate user interaction, at each iteration we chose a phrase of up to three tokens from the reference translation which does not appear in the current MT hypotheses. In the *strict* setting, the complete phrase must be missing from the hypothesis. In the *relaxed* setting, only the first word must be missing. Table 1 shows results for a simulated editing session with four cycles. When a three-token phrase cannot be found, we backoff to two-token phrases, then to single tokens as constraints. If a hypothesis already matches the reference, no constraints are added. By specifying a new constraint of up to three words at each cycle, an increase of over 20 BLEU points is achieved in all language pairs.

### 4.2 Domain Adaptation via Terminology

The requirement for use of domain-specific terminologies is common in real-world applications of MT (Crego et al., 2016). Existing approaches incorporate placeholder tokens into NMT systems, which requires modifying the pre- and post- processing of the data, and training the system with

---

[2]NMT models do not use explicit alignment between source and target, so we cannot use alignment information to map target phrases to source phrases

data that contains the same placeholders which occur in the test data (Crego et al., 2016). The MT system also loses any possibility to model the tokens in the terminology, since they are represented by abstract tokens such as "⟨TERM_1⟩". An attractive alternative is to simply provide term mappings as constraints, allowing any existing system to adapt to the terminology used in a new test domain.

For the target domain data, we use the Autodesk Post-Editing corpus (Zhechev, 2012), which is a dataset collected from actual MT post-editing sessions. The corpus is focused upon software localization, a domain which is likely to be very different from the WMT data used to train our general domain models. We divide the corpus into approximately 100,000 training sentences, and 1000 test segments, and automatically generate a terminology by computing the Pointwise Mutual Information (PMI) (Church and Hanks, 1990) between source and target n-grams in the training set. We extract all n-grams from length 2-5 as terminology candidates.

$$\mathbf{pmi}(\mathbf{x}; \mathbf{y}) = \log \frac{p(x, y)}{p(x)p(y)} \qquad (5)$$

$$\mathbf{npmi}(\mathbf{x}; \mathbf{y}) = \frac{\mathbf{pmi}(\mathbf{x}; \mathbf{y})}{\mathbf{h}(\mathbf{x}, \mathbf{y})} \qquad (6)$$

Equations 5 and 6 show how we compute the normalized PMI for a terminology candidate pair. The PMI score is normalized to the range $[-1, +1]$ by dividing by the entropy $\mathbf{h}$ of the joint probability $\mathbf{p}(\mathbf{x}, \mathbf{y})$. We then filter the candidates to only include pairs whose PMI is $\geq 0.9$, and where both the source and target phrases occur at least five times in the corpus. When source phrases that match the terminology are observed in the test data, the corresponding target phrase is added to the constraints for that segment. Results are shown in Table 2.

This simple method of domain adaptation leads to a significant improvement in the BLEU score of the model without any human intervention. Thus, manually created domain terminologies are likely to lead to even greater performance gains. Surprisingly, even an automatically created terminology combined with the constrained decoding search yields performance improvements of approximately +2 BLEU points for En-De and En-Fr, and a gain of almost 14 points for En-Pt. The large improvement for En-Pt is probably due to

| ITERATION | 0 | 1 | 2 | 3 |
|---|---|---|---|---|
| **Strict Constraints** | | | | |
| **EN-DE** | 18.44 | 27.64 (+9.20) | 36.66 (+9.01) | **43.92** (+7.26) |
| **EN-FR** | 28.07 | 36.71 (+8.64) | 44.84 (+8.13) | **45.48** +(0.63) |
| **EN-PT*** | 15.41 | 23.54 (+8.25) | 31.14 (+7.60) | **35.89** (+4.75) |
| **Relaxed Constraints** | | | | |
| **EN-DE** | 18.44 | 26.43 (+7.98) | 34.48 (+8.04) | **41.82** (+7.34) |
| **EN-FR** | 28.07 | 33.8 (+5.72) | 40.33 (+6.53) | **47.0** (+6.67) |
| **EN-PT*** | 15.41 | 23.22 (+7.80) | 33.82 (+10.6) | **40.75** (+6.93) |

Table 1: Results for four simulated editing cycles using WMT test data. EN-DE uses *newstest2013*, EN-FR uses *newstest2014*, and EN-PT uses the Autodesk corpus discussed in Section 4.2. Improvement in BLEU score over the previous cycle is shown in parentheses. * indicates use of our test corpus created from Autodesk post-editing data.

the training data for this system being very different from the IT domain (see Appendix). Using a terminology with GBS is likely to benefit any model where the terminology used in the test domain is significantly different from the original training data.

| System | BLEU |
|---|---|
| **EN-DE** | |
| Baseline | 26.17 |
| PMI Ngram Constraints | **27.99** (+1.82) |
| **EN-FR** | |
| Baseline | 32.45 |
| PMI Ngram Constraints | **35.05** (+2.59) |
| **EN-PT** | |
| Baseline | 15.41 |
| PMI Ngram Constraints | **29.15** (+13.73) |

Table 2: BLEU Results for EN-DE, EN-FR, and EN-PT terminology experiments using the Autodesk Post-Editing Corpus

### 4.3 Analysis

Subjective analysis of decoder output shows that phrases added as constraints are not only placed correctly within the output sequence, but also have global effects upon translation quality. This is a desirable effect for user interaction, since it implies that users can bootstrap quality by adding the most critical constraints (i.e. those that are most essential to the output), first. Table 3 shows several examples from the experiments in Table 1, where the addition of lexical constraints was able to guide our NMT systems away from initially quite low-scoring hypotheses to outputs which perfectly match the reference translations.

## 5 Related Work

Most related work to date has presented modifications of SMT systems for specific usecases which constrain MT output via auxilliary inputs. The largest body of work considers **Interactive Machine Translation** (IMT): an MT system searches for the optimal target-language suffix given a complete source sentence and a desired prefix for the target output (Foster, 2002; Barrachina et al., 2009; Green, 2014). IMT can be viewed as subcase of constrained decoding, where there is only one constraint which is guaranteed to be placed at the beginning of the output sequence. Wuebker et al. (2016) introduce **prefix-decoding**, which modifies the SMT beam search to first ensure that the *target prefix* is covered, and only then continues to build hypotheses for the suffix using beams organized by coverage of the remaining phrases in the source segment.

Recently, some attention has also been given to SMT decoding with multiple lexical constraints. The Pick-Revise (PRIMT) (Cheng et al., 2016) framework for Interactive Post Editing introduces the concept of *edit cycles*. Translators specify constraints by editing a part of the MT output that is incorrect, and then asking the system for a new hypothesis, which must contain the user-provided correction. This process is repeated, maintaining constraints from previous iterations and adding new ones as needed. Importantly, their approach relies upon the phrase segmentation provided by the SMT system. The decoding algorithm can only make use of constraints that match phrase

| EN-DE | |
| --- | --- |
| **Source** | |
| He was also an anti- smoking activist and took part in several campaigns . | |
| **Original Hypothesis** | |
| Es war auch ein Anti- Rauch- Aktiv- ist und nahmen an mehreren Kampagnen teil . | |
| **Reference** | **Constraints** |
| Ebenso setzte er sich gegen das Rauchen ein und nahm an mehreren Kampagnen teil . | (1) Ebenso setzte er |
| **Constrained Hypothesis** | (2) gegen das Rauchen |
| **Ebenso setzte er** sich **gegen das Rauchen** ein und **nahm** an mehreren Kampagnen teil . | (3) nahm |

| EN-FR | |
| --- | --- |
| **Source** | |
| At that point I was no longer afraid of him and I was able to love him . | |
| **Original Hypothesis** | |
| Je n'avais plus peur de lui et j'ètais capable de l'aimer . | |
| **Reference** | **Constraints** |
| Lá je n'ai plus eu peur de lui et j'ai pu l'aimer . | (1) Lá je n'ai |
| **Constrained Hypothesis** | (2) j'ai pu |
| **Lá je n'ai** plus **eu** peur de lui et **j'ai pu** l'aimer . | (3) eu |

| EN-PT | |
| --- | --- |
| **Source** | |
| Mo- dif- y drain- age features by selecting them individually . | |
| **Original Hypothesis** | |
| - Já temos as características de extracção de idade , com eles individualmente . | |
| **Reference** | **Constraints** |
| Modi- fique os recursos de drenagem ao selec- ion- á-los individualmente . | (1) drenagem ao selec- |
| **Constrained Hypothesis** | (2) Modi- fique os |
| **Modi- fique os recursos** de **drenagem ao selec-** ion- á-los individualmente . | (3) recursos |

Table 3: Manual analysis of examples from lexically constrained decoding experiments. "-" followed by whitespace indicates the internal segmentation of the translation model (see Section 3.2)

boundaries, because constraints are implemented as "rules" enforcing that source phrases must be translated as the aligned target phrases that have been selected as constraints. In contrast, our approach decodes at the token level, and is not dependent upon any explicit structure in the underlying model.

Domingo et al. (2016) also consider an interactive scenario where users first choose portions of an MT hypothesis to keep, then query for an updated translation which preserves these portions. The MT system decodes the source phrases which are not aligned to the user-selected phrases until the source sentence is fully covered. This approach is similar to the system of Cheng et al., and uses the "forced decoding" feature in Moses (Koehn et al., 2007).

Some recent work considers the inclusion of soft lexical constraints directly into deep models for dialog generation, and special cases, such as recipe generation from a list of ingredients (Wen et al., 2015; Kiddon et al., 2016). Such constraint-aware models are complementary to our work, and could be used with GBS decoding without any change to the underlying models.

To the best of our knowledge, ours is the first work which considers general lexically constrained decoding for any model which outputs sequences, without relying upon alignments between input and output, and without using a search

organized by coverage of the input.

## 6 Conclusion

Lexically constrained decoding is a flexible way to incorporate arbitrary subsequences into the output of any model that generates output sequences token-by-token. A wide spectrum of popular text generation models have this characteristic, and GBS should be straightforward to use with any model that already uses beam search.

In translation interfaces where translators can provide corrections to an existing hypothesis, these user inputs can be used as constraints, generating a new output each time a user fixes an error. By simulating this scenario, we have shown that such a workflow can provide a large improvement in translation quality at each iteration.

By using a domain-specific terminology to generate target-side constraints, we have shown that a general domain model can be adapted to a new domain without any retraining. Surprisingly, this simple method can lead to significant performance gains, even when the terminology is created automatically.

In future work, we hope to evaluate GBS with models outside of MT, such as automatic summarization, image captioning or dialog generation. We also hope to introduce new constraint-aware models, for example via secondary attention mechanisms over lexical constraints.

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

## A   NMT System Configurations

We train all systems for 500000 iterations, with validation every 5000 steps. The best single model from validation is used in all of the experiments for a language pair. We use $\ell_2$ regularization on all parameters with $\alpha = 1e^{-5}$. Dropout is used on the output layers with $p(drop) = 0.5$. We sort mini-batches by source sentence length, and reshuffle training data after each epoch.

All systems use a bidirectional GRUs (Cho et al., 2014) to create the source representation and GRUs for the decoder transition. We use AdaDelta (Zeiler, 2012) to update gradients, and clip large gradients to 1.0.

| Training Configurations | |
|---|---|
| **EN-DE** | |
| Embedding Size | 300 |
| Recurrent Layers Size | 1000 |
| Source Vocab Size | 80000 |
| Target Vocab Size | 90000 |
| Batch Size | 50 |
| **EN-FR** | |
| Embedding Size | 300 |
| Recurrent Layers Size | 1000 |
| Source Vocab Size | 66000 |
| Target Vocab Size | 74000 |
| Batch Size | 40 |
| **EN-PT** | |
| Embedding Size | 200 |
| Recurrent Layers Size | 800 |
| Source Vocab Size | 60000 |
| Target Vocab Size | 74000 |
| Batch Size | 40 |

### A.1   English-German

Our English-German training corpus consists of 4.4 Million segments from the Europarl (Bojar et al., 2015) and CommonCrawl (Smith et al., 2013) corpora.

### A.2   English-French

Our English-French training corpus consists of 4.9 Million segments from the Europarl and CommonCrawl corpora.

### A.3   English-Portuguese

Our English-Portuguese training corpus consists of 28.5 Million segments from the Europarl, JRC-Aquis (Steinberger et al., 2006) and OpenSubtitles[3] corpora.

---

[3]http://www.opensubtitles.org/

