# Peer review of "Lexically Constrained Decoding for Sequence Generation Using Grid Beam Search"

_ACL 2017 — decision unknown_

[Official Review · Reviewer 1 · rating 4 · confidence 4]
soundness 4 · originality 4 · clarity 5 · impact 4 · substance 3 · appropriateness 5 · meaningful comparison 4 · presentation format Poster

This paper describes a straightforward extension to left-to-right beam search
in order to allow it to incorporate lexical constraints in the form of word
sequences that must appear in MT output. This algorithm is shown to be
effective for interactive translation and domain adaptation.

Although the proposed extension is very simple, I think the paper makes a
useful contribution by formalizing it. It is also interesting to know that NMT
copes well with a set of unordered constraints having no associated alignment
information. There seem to be potential applications for this technique beyond
the ones investigated here, for example improving NMT’s ability to handle
non-compositional constructions, which is one of the few areas where it still
might lag traditional SMT.

The main weakness of the paper is that the experiments are somewhat limited.
The interactive MT simulation shows that the method basically works, but it is
difficult to get a sense of how well - for instance, in how many cases the
constraint was incorporated in an acceptable manner (the large BLEU score
increases are only indirect evidence). Similarly, adaptation should have been 
compared to the standard “fine-tuning” baseline, which would be relatively
inexpensive to run on the 100K Autodesk corpus.

Despite this weakness, I think this is a decent contribution that deserves to
be published.

Further details:

422 Given its common usage in PBMT, “coverage vector” is a potentially
misleading term. The appropriate data structure seems more likely to be a
coverage set.

Table 2 should also give some indication of the number of constraints per
source sentence in the test corpora, to allow for calibration of the BLEU
gains.